# Development of an Electrochemical CCL5 Chemokine Immunoplatform for Rapid Diagnosis of Multiple Sclerosis

**DOI:** 10.3390/bios12080610

**Published:** 2022-08-07

**Authors:** Sara Guerrero, Esther Sánchez-Tirado, Lourdes Agüí, Araceli González-Cortés, Paloma Yáñez-Sedeño, José M. Pingarrón

**Affiliations:** Department of Analytical Chemistry, Faculty of Chemistry, University Complutense of Madrid, 28040 Madrid, Spain

**Keywords:** electrochemical immunosensor, multiple sclerosis, CCL5 chemokine, serum, cytokines

## Abstract

Serum level of CCL5 chemokine is considered an emerging biomarker for multiple sclerosis (MS). Due to the lack of specific assays for this disease, the development of a point-of-care test for rapid detection of MS could lead to avoiding diagnostics delays. In this paper, we report the first electrochemical immunoplatform for quantification of the CCL5 biomarker at the clinically required levels, able to discriminate between patients diagnosed with MS and healthy individuals. The immunosensing device involves protein capture from biological samples by complexation with biotinylated specific antibodies immobilized onto neutravidin-functionalized microparticles and sandwich assay with anti-CCL5 antibody and IgG labelled with horseradish peroxidase (HRP) for the enzyme-catalyzed amperometric detection of H_2_O_2_ using hydroquinone (HQ) as the redox mediator. The method shows excellent analytical performance for clinical application with a wide linear range of concentrations (0.1–300 ng·mL^−1^ CCL5, R^2^ = 0.998) and a low detection limit (40 pg·mL^−1^ CCL5). The biosensing platform was applied to the determination of the CCL5 endogenous content in 100-fold diluted sera both from healthy individuals and patients diagnosed with MS, with no further sample treatment in just two hours. The results were successfully compared with those obtained by the ELISA methodology.

## 1. Introduction

Cytokines play an increasingly relevant role as biomarkers in the early diagnosis and monitoring of several diseases. Particularly, chemotactic cytokines (or chemokines) actively participate in the transmigration of various cell types through the blood–brain barrier (BBB) at sites of tissue damage. As an example, the activity of β-chemokine ligand 5 (CCL5) also known as RANTES (regulated on activation of normal T cell expressed and secreted), is binding to chemokine receptors (CCRs) located on immune cells promoting their migration by chemotaxis. It has been proposed that there is a relationship between the CCL5 expression and cellular activities connected to medical conditions. Indeed, serum levels of this chemokine have been suggested as a biomarker for various predominantly inflammatory pathologies, such as rheumatoid arthritis [1], atopic dermatitis or asthma [2], and multiple sclerosis (MS) [3].

Multiple sclerosis (MS) is a neuroinflammatory and degenerative disease in which many cytokines are released in the cerebrospinal fluid (CSF) affecting neuronal excitability [4]. It has been shown that the inflammatory process typical of MS is associated with an increase in the levels of CCL5, which causes in turn significant changes in synaptic excitability of central neurons [5]. Recent reports have proven the link of CCL5 protein to MS development and the CCL5 triggering of inflammation in the central nervous system, which might be key in the disease’s development [6]. Moreover, there are data about the remarkably increased levels of CCL5 associated with MS activity. Various authors have found a higher level of CCL5 in serum and CSF for both the MS groups and the patients with no symptoms of inflammatory–non-inflammatory neurological disease (NIND) during a relapse in MS [4,7,8,9]. The rise in the CCL5 level during the attack has been found that is connected to an increase in the activity of Th1 cells and the deregulation of the cytokine network by the activation of the monocyte/macrophage type of cells [10]. In serum, a standard level of 29 ng·mL^−1^ CCL5 for healthy individuals has been reported [11,12] while it can increase up to a mean of 65 ng mL^−1^ in MS patients [6] although even larger concentrations, 120 ng mL^−1^ CCL5, have been claimed [13]. In general, levels of CCL5 were markedly elevated in the progressive forms of MS with respect to healthy controls. Thus, high circulating levels of the chemokine might be reflecting a more progressive and disabling disease course [14].

Despite the prevalence and severity of MS, the lack of diagnostic tests causes delays in the disease treatment. Indeed, there is not any immunosensor reported to date which allows the selective quantification of CCL5 at the cut-off level in serum of patients and with the ability to discriminate between healthy individuals and patients with MS disease. Current assays for the determination of this biomarker mainly consist of colourimetric ELISA tests whose commercially available kits provide non-linear calibration plots, most of them requiring long test times, lasting three to almost five hours, and relatively large sample volumes (100 μL). These data highlight the need to develop fast, sensitive and accurate biodetection devices, such as those based on the use of electrochemical transduction due to their excellent analytical properties [15].

We have reported electrochemical immunosensors involving several immobilization strategies for the determination of other cytokines such as interleukin 1β (IL-1β), tumour necrosis factor α (TNF-α) [16], interferon γ (IFN-γ) [17] or chemokine (C-X-C motif) ligand 7 (CXCL7) [18]. All these immunoplatforms involved the integration of the immunoconjugates on the electrode surface and provided rapid, sensitive and accurate bioanalytical tools for clinical applications. However, crucial aspects to assess for ensuring the successful application of immunosensors to complex clinical specimens are the overcoming of non-specific adsorption of biomolecules and the minimization of matrix effects. These challenges can be addressed by using functionalized magnetic microparticles (MBs) as substrates to allow the preparation of the corresponding immunoconjugates outside the sample, thus avoiding undesirable matrix components effects whilst reducing the number of involved steps in the working protocol and facilitating also the incubation and cleaning operations [19].

In order to provide a useful electrochemical immunodetection tool for the analysis of real (not spiked) clinical samples from MS patients, we report in this paper the first immunosensing platform for the determination of CCL5. The immunoplatform involves the use of neutravidin-functionalized magnetic microparticles (Neu-MBs) modified with a biotinylated antibody (anti-CCL5-Biotin) for the specific capture of CCL5 from serum samples. The Neu-MBs exhibit important advantages compared to the frequently used carboxylated MBs [20]. It must be remarked that there is no need for carboxyl group activation by EDC/NHSS chemistry, which allows to save reagents and time (at least 30 min). In addition, the biotin-neutravidin bond is very stable [21], and since neutravidin is a deglycosylated form of avidin, the bonds to lectins are practically undetectable [22]. Moreover, neutravidin is approximately neutrally charged, which minimizes non-specific interactions [23].

A sandwich-type immunoassay was implemented by conjugation of anti-CCL5-biotin-Neu-MBs with a secondary anti-CCL5 antibody and a horseradish peroxidase (HRP) labelled IgG for the enzyme-catalyzed amperometric detection of H_2_O_2_ using hydroquinone (HQ) as the redox mediator, thus resulting in a cathodic current whose variation was directly proportional to the logarithm of CCL5 concentration. As a novelty with respect to other immunosensing devices involving magnetic microparticles, we have performed characterization studies by electrochemical impedance electrochemistry (EIS) through the different steps involved in the preparation of the immunoconjugates. 

The amperometric method showed excellent analytical performance with a wide linear range of concentrations (0.1–300 ng·mL^−1^ CCL5, R^2^ = 0.998) and a low detection limit (40 pg·mL^−1^ CCL5). The stability and selectivity of biotin-neutravidin binding along with the sensitivity of amperometric transduction at disposable screen-printed carbon electrodes (SPCEs) provided competitive advantages, in terms of assay time and reproducibility, against commercially available ELISA kits. Moreover, the obtained analytical characteristics fulfilled the requirements for application to real clinical samples, as was demonstrated by determining the CCL5 endogenous content in 100-fold diluted sera from healthy individuals and patients diagnosed with MS, with no further sample treatment in an assay time of two hours.

## 2. Experimental

### 2.1. Apparatus and Electrodes

A CHI1030B (CH Instruments, Inc., Austin, TX, USA) potentiostat controlled by the CHI1030B software was used for the amperometric measurements. Impedimetric measurements were performed with a μAutolab type III potentiostat controlled by FRA2 (EcoChemie, Utrecht, The Netherlands) software. Screen-printed carbon electrodes (SPCEs, DRP110, 4-mm Ø) and a specific cable connector (DRP-CAC) from Metrohm-DropSens (Oviedo, Spain) were used. The measurements were made at room temperature in stirred solutions using 10 mL glass electrochemical cells from Pobel, where the SPCEs were placed on a homemade polymethylmethacrylate (PMMA) casing with an embedded neodymium magnet (AIMAN GZ), were immersed. A DynaMag-2 magnetic concentrator (Cat. No. 12321D, Dynabeads^®^, Invitrogen-Thermofisher Scientific, Waltham, MA, USA), a Bunsen AGT-9 Vortex, a Thermo-Shaker MT100 (Universal Labortechnik, Leipzig, Germany), a P-Selecta (Cencom, Barcelona, Spain) centrifuge, a Crison model Basic 20+ pH-meter, an Elmasonic S-60 (Elma) ultrasonic bath and a magnetic stirrer MS01 (ELMI, Ltd., Riga, Latvia) were employed. ELISA absorbance readings were made in a Sunrise™ Tecan microplate reader with the Magellan V 7.1 software (Männedorf, Switzerland).

### 2.2. Reagents and Solutions

A biotinylated goat anti-human CCL5/RANTES antibody (anti-CCL5-Biotin, R&D Systems, Minneapolis, MN, USA, Catalog Number BAF478) and a mouse anti-human CCL5/RANTES antibody (anti-CCL5) were used as the capture and secondary antibodies, respectively. A recombinant human CCL5/RANTES standard (CCL5) was used as the antigen. These last immunoreagents were from the DuoSet ELISA Human CCL5/RANTES (R&D Systems, Cat. No. DY278). Reagent Diluent from the same ELISA kit consisted of 1% BSA in saline phosphate buffer (PBS) of pH 7.4 and was used for reconstituting both the capture antibody and the antigen. An HRP-labelled goat anti-mouse IgG Fc detection antibody (HRP-IgG, Abcam, Cambridge, UK, Cat. No. ab97265) was also used.

A phosphate buffer saline solution (PBS) of pH 7.4 consisted of 137 mM NaCl (Labkem), 2.7 mM KCl (Scharlab), 8.1 mM Na_2_HPO_4_, 1.5 mM KH_2_PO_4_, a 50 mM potassium phosphate buffer of pH 6.0 (PB), and a PBST buffer solution prepared by addition of 0.05% Tween 20 to PBS, were used. Deionized water was from a Millipore Milli-Q purification system (18.2 MΩ cm). Sera from healthy individuals and from MS patients were purchased from Central BioHub^®^ Specimens (Central BioHub GmbH, Hennigsdorf, Germany). All samples were stored at −80 °C until their use. Neutravidin-functionalized magnetic microparticles (Neu-MBs, 1 μm-Ø, 10 mg mL^−^^1^, SpeedBeads™) were from GE Healthcare, Chicago, IL, USA. Hydroquinone (HQ) and hydrogen peroxide (H_2_O_2_) (30% *v*/*v*) was provided by Sigma-Aldrich, San Luis, AZ, USA.

Haemoglobin (HB, Sigma-Aldrich, San Luis, AZ, USA, Cat. No. H7379), chemokine (C-X-C motif) ligand 7 (CXCL7, R&D Systems Inc., Minneapolis, MN, USA, DY393 ELISA kit), transforming growth factor β1 (TGFβ1, R&D Systems Inc., Minneapolis, MN, USA, DY240-05), interleukin-1β (IL-1β, R&D Systems, Minneapolis, MN, USA, DY201-05), interleukin 6 (IL-6, Abcam, Cambridge, UK, K130627), matrix metalloproteinase 3 (MMP3, R&D Systems Inc., Minneapolis, MN, USA, Ref. 841045), rheumatoid factor (RF, Abcam, Cambridge, UK, 178653 ELISA kit), endoglin (CD105, R&D Systems Inc., Minneapolis, USA, DY1097 ELISA kit), human serum albumin (HSA, Sigma-Aldrich, San Luis, USA, Cat. No. A1653), tumour necrosis factor α (TNFα, R&D Systems Inc., Minneapolis, USA, DY210-05), anti-cyclic citrullinated peptide antibody (CCPA, Demeditec, Kiel, Germany, DE7760 ELISA kit), human immunoglobulin (hIgG, Sigma-Aldrich, San Luis, USA, Ref. I2511) were tested as potential interfering compounds.

### 2.3. Preparation of the Magnetic Bioconjugates

A volume of 3 µL of the Neu-MBs suspension was transferred to a 1.5 mL Eppendorf tube and two washings were made with 50 µL PBST. Then, the particles were separated by means of the magnetic separator, the supernatant discarded, and 25 µL of a 25 µg mL^−1^ anti-CCL5-Biotin in PBS was added, allowing incubation for 30 min at 25 °C under continuous stirring at 600 rpm. Next, two washings were carried out with 50 µL PBST, magnetized again, the supernatant discarded, and 25 µL of the standard CCL5 solution (or the sample) was added, incubating for 60 min at 25 C and washing twice with 50 µL PBST. Subsequently, 25 µL of a 0.5 µg mL^−1^ anti-CCL5 solution in PBS was added and the anti-CCL5-CCL5-anti-CCL5-Biotin-Neu-MBs conjugates were incubated for 30 min at 25 °C, under stirring at 600 rpm and washed again as in the previous step. Finally, 25 µL of 0.5 µg mL^−1^ HRP-IgG in PBS was added and incubated for 30 min at 25 °C under continuous stirring at 600 rpm and washed twice with 50 µL PBST.

### 2.4. Amperometric Detection

Once the SPCE was placed on the homemade PMMA casing provided with the embedded neodymium magnet, the HRP-IgG-anti-CCL5-CCL5-anti-CCL5-Biotin-Neu-MBs (50 μL) were cast onto the working electrode surface. Then, the potentiostat was connected, the casing–electrode assembly immersed into the electrochemical cell containing 10 mL of 0.05 mM PB pH 6.0 and 100 µL of freshly prepared 0.1 mol L^−1^ HQ solution, and a constant potential of −0.20 V vs. Ag pseudo-reference electrode was applied. The amperometric current was recorded under stirring. Upon stabilization of the background current (approximately 100 s), 50 µL of a 0.1 mol L^−1^ H_2_O_2_ solution freshly prepared in PB, pH 6.0, were added to the cell, and the cathodic current arising from the electrochemical reduction of the quinone formed in the enzymatic reduction of H_2_O_2_ mediated by HQ was recorded until the steady state was reached. Amperometric signals were calculated as the difference between such steady-state currents and the background currents. The given values are the mean of three replicates and the error bars were estimated as three times the standard deviation of each set of replicates (α = 0.05).

### 2.5. Analysis of Serum Samples

The developed immunosensor was applied to the determination of CCL5 in serum samples. Analyzed sera from healthy individuals and MS patients were purchased from Central BioHub respecting all the ethical issues and relevant guidelines and regulations existing for these samples. Sera were stored at −80 °C until use. Once the absence of matrix effect was verified for 100 times diluted samples, the determination of CCL5 was performed by applying the procedures described in Section 2.3 and Section 2.4 to 25 μL of diluted serum in PBS pH 7.4, through simple interpolation of the signals obtained for the samples into the calibration graph constructed with the CCL5 standards. The obtained results were compared with those provided by an ELISA kit using anti-CCL5 pre-coated wells and sandwich immunoassay with anti-CCL5-Biotin and HRP-Streptavidin following the protocol recommended in the kit www.rndsystems.com/products/mouse-ccl5-rantes-duoset-elisa_dy478#product-datasheets (accesed on 6 August 2022).

## 3. Results and Discussion

Figure 1 shows schematically the different steps involved in the preparation and functioning of the HRP-IgG-anti-CCL5-CCL5-anti-CCL5-Biotin-Neu-MBs/SPCE immunosensor. As indicated in the Experimental section, anti-CCL5-Biotin capture antibodies were immobilized onto Neu-MBs and thereafter, target CCL5 was sandwiched with an anti-CCL5 secondary antibody followed by conjugation with HRP-IgG. Once the SPCE was placed on the PMMA casing, the magnetic immunoconjugate was cast onto the working electrode surface and the prepared immunosensor was immersed in 10 mL of 50 mM PB solution of pH 6.0 containing 1 mM HQ. Then, the amperometric responses were recorded after the addition of H_2_O_2_ by measuring the cathodic current at −0.20 V (vs. the Ag pseudo-reference electrode) according to the displayed reactions.

### 3.1. Optimization of the Variables Involved in the Immunosensor Performance

The influence of the experimental variables affecting the measurements of the immunosensor for CCL5 was evaluated. Considering the proposed sandwich-type configuration, the larger ratio between the currents measured with the immunosensor in the presence of a 2 ng·mL^−1^ CCL5 standard (S) and in its absence (N), S/N ratio, was adopted as the optimization criterion. The tested variables were: loading of anti-CCL5-Biotin capture antibody onto Neu-MBs and incubation time; loading of anti-CCL5 secondary antibody onto CCL5-anti-CCL5-Biotin-Neu-MBs and incubation time; and loading of HRP-IgG onto anti-CCL5-CCL5-anti-CCL5-Biotin-Neu-MBs and incubation time. Other experimental conditions involved in the immunosensor performance such as the amount of MBs, the applied potential (−0.20 V vs. Ag pseudo-reference electrode) or the concentration of the H_2_O_2_/HQ system were taken from previous works [24,25]. The obtained results are displayed in Figure 2 and summarized in Table 1.

Figure 2a shows the results obtained in the optimization of the anti-CCL5-Biotin loading onto Neu-MBs. The specific amperometric current (grey bars) increased with the amount of capture antibody over the 10–25 μg·mL^−1^ range and decreased for larger concentrations, probably due to the sterically hindered binding to Neu-MBs with a large loading of anti-CCL5-Biotin. Furthermore, the unspecific responses (white bars) measured in absence of CCL5 remained practically constant over the whole tested range. Accordingly, a larger specific-to-unspecific current ratio (S/N) was observed for 25 μg·mL^−1^ anti-CCL5-Biotin which was selected for further work. The incubation time of anti-CCL5-Biotin onto Neu-MBs was checked over the 15 to 90 min range with the results displayed in Figure 2b. As it can be observed, 30 min was sufficient to allow an efficient immobilization of the capture antibody and therefore for the preparation of the immunosensor.

The effect of the anti-CCL5 secondary antibody loading on the immunosensor response was tested over the 0.25 to 1.0 μg·mL^−1^ range (Figure 2c). A larger S/N ratio was found for a 0.5 μg·mL^−1^ concentration. Furthermore, Figure 2d shows as the unspecific currents measured in the absence of CCL5 remained practically constant over the whole range of incubation times tested. Therefore, according to the larger value of the specific-to-unspecific current ratio, 30 min was selected as the incubation time of anti-CCL5 onto CCL5-anti-CCL5-Biotin-Neu-MBs for further work.

Figure 2e shows a behaviour of the amperometric responses measured with different HRP-IgG loadings similar to that observed for the secondary anti-CCL5 antibody concentration. A larger S/N ratio was found for 0.5 μg·mL^−1^ which was the HRP-IgG concentration used for the preparation of the immunosensor. Regarding the incubation time for HRP-IgG (Figure 2f), 30 min was sufficient to achieve a practically constant specific signal and provide a larger S/N ratio.

The results of these optimization studies are summarized in Table 1, which includes the tested ranges and the selected values for each variable. It is worth mentioning the excellent S/N ratio in the obtained data set, with a specific to non-specific ratio (S/N) of approximately 16 at the optimized experimental conditions. The significantly low background currents are probably due to the application of the magnetically assisted methodology and, in particular, to the use of Neu-MBs instead of streptavidin- or avidin-functionalized MBs. Indeed, as stated above, the practically undetectable unions to lectin and the nearly neutral charge of neutravidin minimize non-specific interactions [24]. Other experimental variables affecting the performance of the immunosensor were those recommended or selected in previous papers. As an example, the incubation time of the target CCL5 onto anti-CCL5-Biotin-Neu-MBs was optimized taking as a reference the value of 120 min used by the ELISA kit with the same reagents www.rndsystems.com/products/human-ccl5-rantes-duoset-elisa_dy278 (accesed on 6 August 2022). However, only 60 min was enough to obtain the best amperometric responses, probably due to the much lower volume of CCL5 solution required for the preparation of the immunoconjugate, 25 μL, a quarter of that used in the ELISA test. Furthermore, the amount of Neu-MBs was that which was previously optimized [24], as well as the applied potential (−0.20 V vs. Ag pseudo-reference electrode) and the composition of the H_2_O_2_/HQ system [25].

### 3.2. Characterization Studies

The different steps involved in the preparation of the magnetic immunoconjugates were characterized by EIS to evaluate the success of the developed procedure. Few articles have reported the use of EIS with functionalized MBs. However, the ability of this technique to investigate the change in resistance and capacitance of the interface providing insights into the effects of incubation of imunoreagents in the charge transfer process at electrodes modified with MBs have been demonstrated in various articles [26,27,28].

In this paper, impedance measurements at SPCEs modified with Neu-MBs were performed by plotting the Nyquist spectra obtained in 5 mM Fe(CN)_6_
^3−/4−^ solutions prepared in 0.1 M phosphate buffer of pH 7. Figure 3a shows the EIS curves at different steps of MBs modification with the immunoreagents recorded at the open circuit in the 10^5^–0.040 Hz frequency range. The bare SPCE exhibits a small semicircle, with a diameter of 868 Ω (dark blue, curve 1) suggesting high electron transfer and low impedance. After magnetic trapping of Neu-MBs, the charge transfer resistance increased to 1212 Ω (light blue, curve 2) indicating that the redox probe Fe(CN)_3_^3−/4−^ ions have reduced ability to participate in the electron transfer due to the presence of the insulating layer composed of polymeric magnetic beads covered with the neutravidin biomolecules. The R_CT_ measured upon formation of anti-CCL5-Biotin-Neu-MBs/SPCE increased to 1470 Ω (yellow, curve 3) most likely due to the attachment of the antibodies hindering the arrival of the redox probe to the electrode surface thus confirming the success in the preparation of the immunoconjugates onto Neu-MBs. However, after incubation of the CCL5 antigen, the immunocomplex formation provoked a strong decrease in the R_CT_ value to 779 Ω (red, curve 4). This effect was also observed for other affinity biosensors involving the use of functionalized MBs as support of bioreagents [26,27]. A possible explanation of this behaviour relies on the value of the CCL5 isoelectric point, 9.27 [29]. This value indicates that CCL5 is positively charged at the working pH and, therefore, the electrostatic attraction to the redox probe makes the resistance to charge transfer lower. In addition, as expected, the R_CT_ values increased significantly to 1004 Ω upon immobilization of the anti-CCL5 secondary antibody (purple, curve 5), and to 1483 Ω after the incorporation of the HRP-IgG conjugate (grey, curve 6). Figure 3a shows also the equivalent circuit proposed after successive incubation steps. The experimental results fit well the Randles circuit depicted at the top of Figure 3a, where Rs is the electrolyte solution resistance in series with the parallel combination of the double-layer capacitance, C_dl_, the charge transfer resistance at the electrode/solution interface, R_CT_, and the Warburg element, Z_W_, semi-infinite impedance due to diffusion from the core of the solution to the interface which is related to the faradaic reaction [30,31].

The obtained results confirmed the successful preparation of the magneto-immunoconjugates using SPCEs modified with magnetically trapped Neu-MBs at the optimized experimental conditions. In addition, these results suggest the possibility of developing an impedance method to determine CCL5 using the Rct values as the analytical signal for calibration. As a proof of concept, several Nyquist curves for different concentrations of CCL5 are shown in Figure 3b, where an increase in the semicircle diameter as the CCL5 concentration increased was observed.

### 3.3. Analytical Characteristics of the Immunosensor for the Determination of CCL5

The calibration plot for CCL5 standards (Figure 4) exhibits a linear dependence between the measured cathodic current and the logarithm of the chemokine concentration over the 0.1–300 ng·mL^−1^ (R^2^ = 0.998) range, according to the equation: i, μA = (1.76 ± 0.02) log ([CCL5], ng·mL^−1^) + (1.95 ± 0.03). The limits of detection (LOD) and quantification (LOQ) were calculated according to the 3s_b_/m and 10s_b_/m criteria, respectively, where s_b_ is the standard deviation of 10 amperometric measurements obtained for 0.0 ng·mL^−1^ CCL5 and m the slope of the calibration plot. The calculated values were 40 and 130 pg·mL^−1^, respectively.

Considering that the expected levels of CCL5 in serum of healthy individuals and patients diagnosed with MS are of some ng·mL^−1^ tens [6,11,12,13], the calibration linear range, extending over three decades of concentration from one-tenth to three hundred ng·mL^−1^, is fully adequate for the determination of this chemokine in serum without the need for applying large sample dilution. Since no biosensors have been reported for the determination of CCL5, the characteristics of the developed method can be compared with those provided by the ELISA immunoassays. Commercial ELISA kits for CCL5 by R&D Systems provide a non-linear double logarithmic calibration plot with a dynamic range from 15.6 to 1000 pg·mL^−1^ using the same immunoreagents as in this work rndsystems.com/products/human-ccl5-rantes-duoset-elisa_dy278 (accesed on 6 August 2022). Interestingly, the use of the immunosensor requires an assay time of 2 h, counting from the preparation of anti-CCL5-Biotin-Neu-MBs, vs. 4 h 45 min needed for ELISA immunoassay. 

The reproducibility of the amperometric responses was checked by preparing five different immunoplatforms on the same day and measuring the currents in the absence and in the presence of 2 ng·mL^−1^ CCL5. The obtained relative standard deviation (RSD) values were 1.5 and 2.3%, respectively. In addition, the responses obtained with five immunosensors prepared on different days provided RSD values of 3.7 and 4.1%, respectively. This reproducibility in the amperometric measurements is much better than that claimed for the ELISA kits, with inter-assay and intra-assay variation coefficient values, CV < 12% and <10%, respectively. An important additional advantage is the small volume of sample required to carry out the determination of CCL5. Indeed, this volume, 25 μL, is four times smaller than that needed with the ELISA kit using the same reagents. Therefore, the analytical features provided by the electrochemical immunoplatform demonstrate the suitability of the immunoplatform for the rapid and accurate determination of CCL5 in clinical samples.

### 3.4. Selectivity

The selectivity of the method was evaluated by comparing the amperometric responses measured for 0 and 2 ng·mL^−1^ CCL5 in the absence and in the presence of various proteins that may coexist with CCL5 in clinical samples. The tested compounds were: haemoglobin (HB), chemokine (C-X-C motif) ligand 7 (CXCL7), transforming growth factor β1 (TGFβ1), interleukin 1β (IL-1β), interleukin 6 (IL-6), matrix metalloproteinase 3 (MMP3), rheumatoid factor (RF), endoglin (CD105), human serum albumin (HSA), factor necrosis tumour α (TNFα), anti-cyclic citrullinated peptide antibody (CCPA), and human immunoglobulin (hIgG). Solutions containing the expected concentration of each compound in the serum of healthy individuals (or higher values) were checked with the results shown in Figure 5.

As it can be observed, no significant interference in the determination of CCL5 was apparent in the presence of the potential interfering compounds at the tested concentration levels. These results evidenced the high specificity of the immunoreagents employed for the recognition of the target protein as well as the excellent selectivity of the amperometric transduction under the optimized working conditions. It should also be noted that the ELISA kit protocol using the same reagents indicated the absence of interference by other proteins such as MCP1 (monocyte chemoattractant protein-1), MIP-α and β (macrophage inflammatory protein), CXCL1 (GRO-α) and CXCL8 (IL8).

### 3.5. Storage Stability

The storage stability of the CCL5 conjugates (stored at 8 °C in Eppendorf tubes containing 25 μL of PBS of pH 7.4) was tested. Various anti-CCL5-Biotin-Neu-MBs immunoconjugates were prepared, stored, and used to prepare the corresponding immunosensors which were employed to measure the amperometric responses for 2 ng·mL^−1^ CCL5 on different days. The results obtained (Figure 6) revealed that the initial amperometric response was maintained within the control limits set at ±2 times the standard deviation of ten measurements carried out on the first day, at least for 25 days after the CCL5-Biotin-Neu-MBs storage (no longer time was tested). This remarkably good storage stability suggests the possibility of preparing a set of anti-CCL5-Biotin-Neu-MBs conjugates and storing them under the above-mentioned conditions until their use for the preparation of the immunosensors.

### 3.6. Determination of CCL5 in Serum of Healthy Individuals and MS Patients

The developed immunoplatform was applied to the determination of CCL5 in serum samples of healthy individuals and patients diagnosed with MS provided by Central BioHub^®^ Specimens (Central BioHub GmbH, Hennigsdorf, Germany). All the ethical issues and relevant guidelines and regulations existing for these samples were accomplished. Sera were stored at −80 °C until use. The possible existence of matrix effects was tested by comparing the calibration plot constructed in serum spiked with the target protein over the 0.1 to 60 ng mL^−1^ concentration range and 100 times diluted with PBS pH 7.4, and the calibration plot constructed with CCL5 standards. The slope value of the resulting linear logarithmic calibration plot (R^2^ = 0.998) was 1.69 ± 0.04 μA per decade of concentration, which is very similar to that of the calibration plot obtained with CCL5 standards (Figure 7a shows the overlapping of both calibration plots over the whole concentration range). A statistical comparison of the slope values by applying the Student’s t-test provided a t_exp_ = 1.467, lower than the tabulated one, t_tab_ = 2.145, thus confirming that no statistically significant differences between the slopes occurred. Therefore, the determination of CCL5 was carried out by interpolation of the amperometric currents measured for the 100-times diluted samples into the calibration plot prepared with standards.

The results obtained by triplicate analysis of serum from healthy individuals and patients diagnosed with MS are summarized in Table 2. Such results are compared with those obtained by using the ELISA kit involving the same reagents. As it can be deduced from the linear correlation plot shown in Figure 7b, with slope and intercept values of 1.0 ± 0.2 and 0.11 ± 0.16, respectively, an excellent agreement between the results obtained with both methods occurred.

Importantly, the obtained results agree with the expected CCL5 concentration ranges reported in the literature [6,11,12,13]. These values show also the expected hyperexpression of this chemokine in the serum of patients with MS. In addition, the difference in concentration with respect to the values found for healthy individuals is sufficiently large (between three times and ten times larger) to allow this biomarker to serve as a good reference for the detection of the disease and the monitoring and follow-up of patients. This large difference between the CCL5 levels of healthy individuals and patients diagnosed with MS is an important advantage versus that found for another target compound related to the immune response in MS [20], the anti-MBP autoantibody. Indeed, the determination of anti-MBP in the same serum samples provided mean concentration levels of the antibody in healthy individuals of 0.50 ± 0.07 ng·mL^−1^ and of 2.26 ± 0.04 ng·mL^−1^ in patients with MS, with a much smaller difference between both data sets [20]. Therefore, we can deduce that the prognosis and monitoring of the disease using anti-MBP as a target biomarker is less accurate than that based on the determination of CCL5.

It is worth mentioning that the differences observed in the concentration of CCL5 for patients with MS are probably due to the different clinical statuses. For instance, it is known that CCL5 levels are significantly larger in obese individuals [31], a circumstance that would explain the high concentration observed in the MS398619 sample which belongs to a woman with a BMI of 32.

## 4. Conclusions

In this work, we report the first amperometric immunosensor developed to date for the determination of the CCL5 chemokine, a reliable biomarker for multiple sclerosis autoimmune disease. The proposed design involves the immobilization of anti-CCL5-Biotin capture antibody onto Neu-MBs for the selective capture of the target protein followed by conjugation with anti-CCL5 and HRP-IgG. The amperometric responses were measured after magnetic capture of HRP-IgG-anti-CCL5-CCL5-anti-CCL5-Biotin-Neu-MBs conjugates onto SPCEs using the H_2_O_2_/HQ system. The developed method shows high sensitivity and a wide linearity range, providing a calibration plot over the 0.1–300 ng·mL^−1^ concentrations with a detection limit of 40 pg·mL. These analytical characteristics, together with the high reproducibility and selectivity, advantageously compare with those of the ELISA methodology since the developed immunoplatform involves a much smaller sample volume and a much shorter assay time. In addition, the developed method fulfils the requirements of applicability to real serum samples allowing the successful discrimination between healthy individuals and MS patients.

## Figures and Tables

**Figure 1 biosensors-12-00610-f001:**
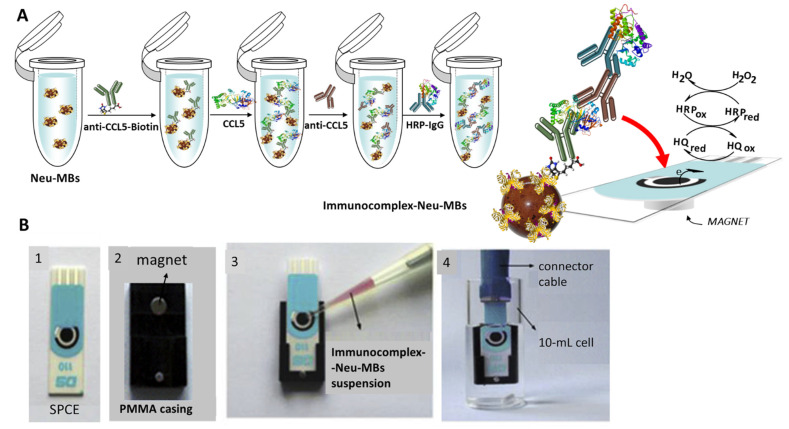
(**A**) Schematic display of the steps involved in the construction of the magnetically assisted im-munoplatform for CCL5 and the reactions providing the amperometric responses. (**B**) Homemade magnet holding block with an encapsulated neodymium magnet (2); addition of the immunocomplex–Neu–MBs suspension onto the electrode surface of SPCE (3); SPCE with 50 μL of the magnetoimmunoconjugate immersed into a 10 mL electrochemical cell to perform the amperometric measurements.

**Figure 2 biosensors-12-00610-f002:**
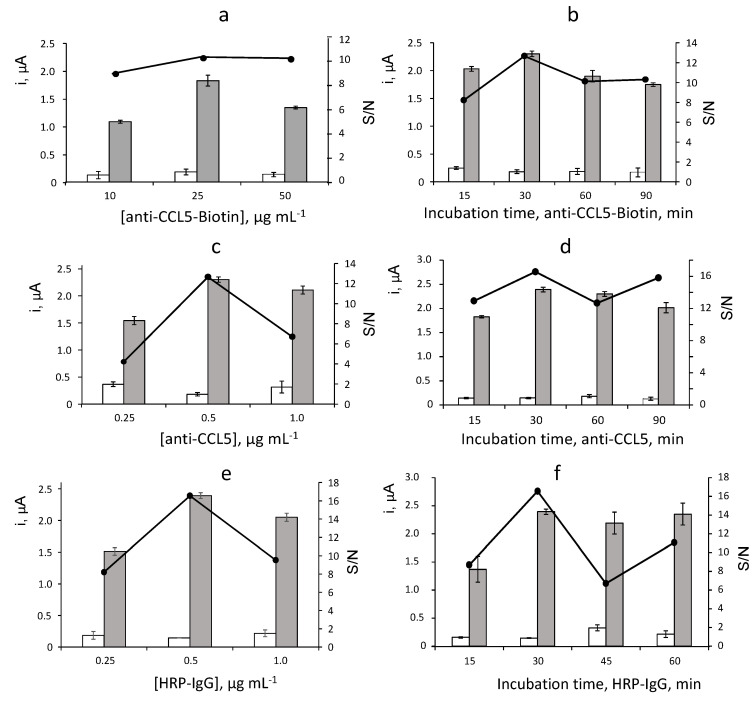
Biotin loading (**a**) and incubation time (**b**) onto Neu–MBs, anti–CCL5 loading (**c**) and incubation time (**d**) onto CCL5–anti–CCL5–Biotin–Neu–MBs, HRP–IgG loading (**e**) and incubation time (**f**) onto anti–CCL5–CCL5–anti–CCL5–Biotin–Neu–MBs, on the amperometric responses of the immunoplatform. 3 μL Neu–MBs; 10–50 μg·mL^−1^ anti–CCL5–Biotin, 60 min (**a**); 25 µg· mL^−1^ anti–CCL5–Biotin, 15–90 min (**b**); 25 μg·mL^−1^ anti–CCL5-Biotin, 30 min (**c**–**f**); 0 (white) or 2 ng· mL^−1^ (grey) CCL5, 60 min; 0.5 μg·mL^−1^ anti–CCL5, 60 min (**a**,**b**); 0.25–1 μg·mL^−1^ anti–CCL5, 60 min (**c**); 0.5 μg·mL^−1^ anti–CCL5, 15–90 min (**d**); 0.25–1 μg·mL^−1^ HRP–IgG, 30 min (**f**); 0.5 μg·mL^−1^ HRP–IgG, 15–60 min (**f**). Error bars estimated as triple of the standard deviation (*n* = 3).

**Figure 3 biosensors-12-00610-f003:**
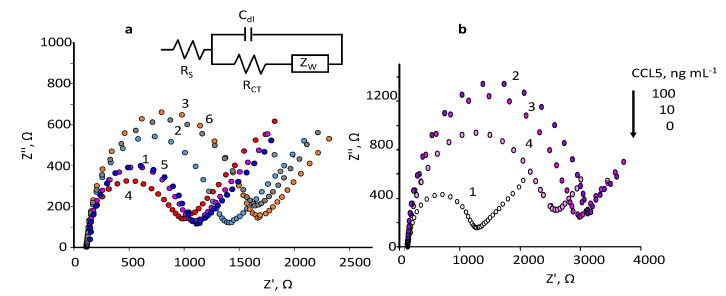
mM Fe(CN)_6_^3^^−^^/4^^−^ in 0.1 M PB of pH 7.4 for: (**a**) bare SPCE (dark blue,1); Neu–MBs/SPCE (light blue, 2); anti–CCL5–Biotin–Neu-MBs/SPCE (yellow, 3); CCL5–anti–CCL5–Biotin–Neu–MBs/SPCE (red, 4); anti–CCL5–CCL5–anti–CCL5–Biotin–Neu–MBs/SPCE (purple, 5); HRP–IgG–CCL5–anti–CCL5–Biotin–Neu–MBs/SPCE (grey, 6). The equivalent circuit used to adjust the experimental results is also shown; (**b**) bare SPCE (white, 1); HRP–IgG–CCL5–anti–CCL5–Biotin–Neu–MBs/SPCE for 100 (violet, 2) 10 (purple, 3) and 0 (pink, 4) ng·mL^−1^ CCL5.

**Figure 4 biosensors-12-00610-f004:**
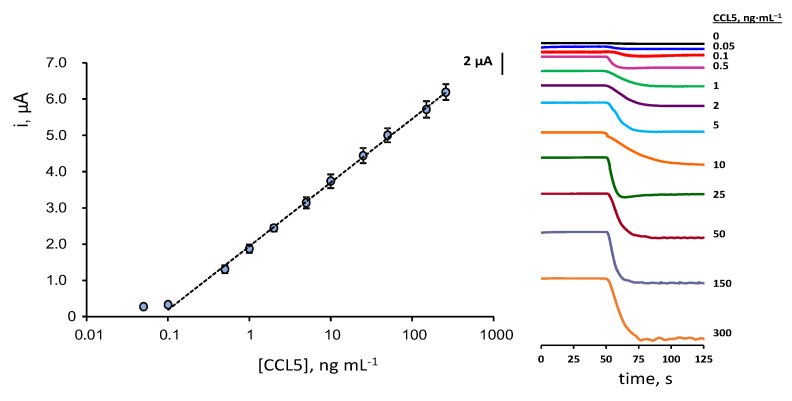
Calibration plot for the determination of CCL5 with the HRP–IgG–anti–CCL5–CCL5–anti–CCL5–Biotin–Neu–MBs/SPCE immunoplatform and real amperometric measurements recorded for different CCL5 concentrations at −0.20 V vs. Ag pseudo-reference electrode. Error bars are estimated as triple the standard deviation (*n* = 3).

**Figure 5 biosensors-12-00610-f005:**
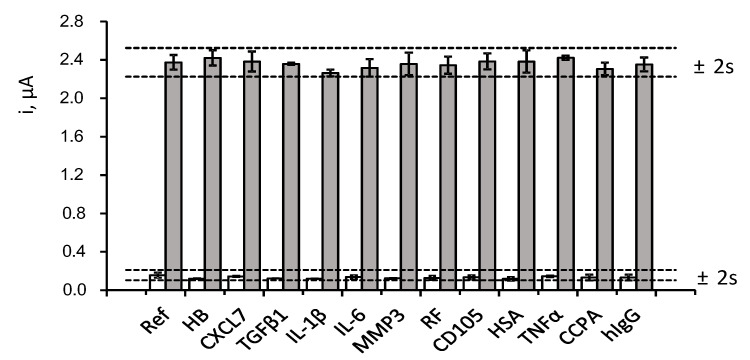
Effect of the presence of 5 mg mL^−1^ HB, 50 ng·mL^−1^ CXCL7, 50 pg·mL^−1^ TGF β1, 50 pg·mL^−1^ IL-1β, 300 pg·mL^−1^ IL-6, 2 ng mL^−1^ MMP3, 100 IU·mL^−1^ RF, 70 pg mL^−1^ CD105, 5 mg·mL^−1^ HSA, 200 pg mL^−1^ TNFα, 100 IU·mL^−1^ CCPA and 1 mg·mL^−1^ hIgG, on the amperometric responses obtained with the developed immunosensor for 0 (white) and 2 ng mL^−1^ CCL5 (grey). Error bars are estimated as triple the standard deviation (*n* = 3).

**Figure 6 biosensors-12-00610-f006:**
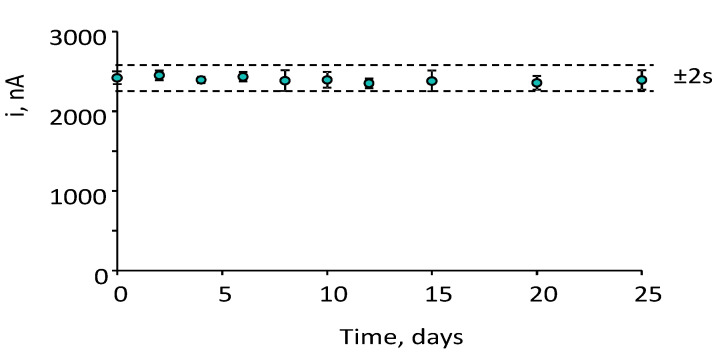
Control chart constructed to check the storage stability of anti–CCL5–Biotin–Neu–MBs conjugates upon storage at 8 °C in Eppendorf tubes containing 25 μL of PBS of pH 7. The central value was the average amperometric signal for 2 ng mL^−1^ CCL5 provided by 10 different immunosensors prepared the day 0 of the study. Upper and lower limits of control were set as twice the standard deviation (2 s) of these measurements. Error bars were estimated as triple the standard deviation (*n* = 3).

**Figure 7 biosensors-12-00610-f007:**
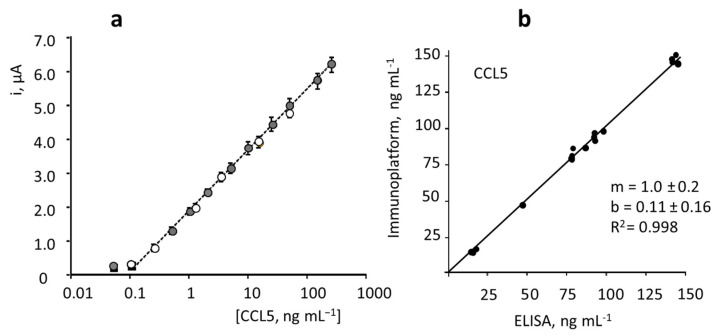
(**a**) Comparison of the calibration plots constructed with the HRP–IgG–anti–CCL5–CCL5–anti–CCL5–Biotin–Neu–MBs/SPCE immunoplatform for CCL5 standards in buffered solutions (white) and in CCL5-spiked100-fold diluted human serum (grey). Error bars estimated as triple standard deviation (*n* = 3). (**b**) Correlation plot between the CCL5 concentrations obtained with the developed immunoplatform and the ELISA method in serum samples (replicates made for each sample are included in the plot).

**Table 1 biosensors-12-00610-t001:** Optimization of the experimental variables involved in the preparation of HRP-IgG-anti-CCL5-CCL5-anti-CCL5-Biotin-Neu-MBs bioconjugates used for the amperometric determination of CCL5.

Variable	Tested Range	Selected Value
anti-CCL5-Biotin, µg·mL^−1^	10–50	25
anti-CCL5-Biotin incubation time, min	15–90	30
anti-CCL5, µg·mL^−1^	0.25–1	0.5
anti-CCL5 incubation time, min	15–90	30
HRP- IgG, µg·mL^−1^	0.25–1	0.5
HRP- IgG incubation time, min	15–60	30

**Table 2 biosensors-12-00610-t002:** Determination of endogenous CCL5 concentration in serum of healthy individuals and MS patients.

Sample	Reference *	CCL5, ng·mL^−1^ Immunosensor ELISA	
Healthy individual	S468087	15.0 ± 0.1	15.5 ± 0.5
Healthy individual	S468160	13 ± 0.1	13.2 ± 0.2
MS patient	MS338745	48.9 ± 0.3	48.6 ± 0.6
MS patient	MS337913	79.7 ± 0.5	80.1 ± 0.5
MS patient	MS354878	94.9 ± 0.5	94.5 ± 0.6
MS patient	MS398619	147.7 ± 0.6	148.1 ± 0.8

Mean values ± ts/√*n*; *n* = 3; α = 0.05. * Reference numbers from the Product List of Central BioHub: CBH#2445 multiple_sclerosis_210621(1).xlsx.

## Data Availability

Not applicable.

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
