# Peer review of "Development of an Electrochemical CCL5 Chemokine Immunoplatform for Rapid Diagnosis of Multiple Sclerosis"

_biosensors, 2022, doi:10.3390/bios12080610_

Round 1

Reviewer 1 Report

This work proposes an electrochemical platform for quantification of the CCL5 chemokine in clinical samples: the experimental approach consists in the immobilization of anti-CCL5-Biotin capture antibody onto neutravidin-functionalized magnetic microparticles for the selective capture of the target protein followed by conjugation with anti-CCL5 and horseradish peroxidase -labelled IgG for the enzyme-catalyzed amperometric detection onto SPCEs using the H2O2/HQ system.

The work is clearly written, well-structured and all claims of the authors are sustained by experimental evidence. Analytical approach is robust. Innovation with respect to literature is clearly reported. Main important result i.e. the detection of CCL5 at the clinically required levels, is clearly highlighted. Ability to discriminate between patients diagnosed with MS and healthy individuals is a further great result.

- In the introduction line 72 maybe a typo occurred: “Bearing this in mind and in to provide”. I think “in order to” should be the correct version - Size of figures and tables should fit the template of the journal - Ohm in figure 3 should be same to the unit (Ω) in main text;; the underline of “8 oC” (line 429) should be deleted.

In conclusion, as it is a very good job, I would recommend this work for publication in Biosensors in the present form.

Author Response

Reviewer 1.

This work proposes an electrochemical platform for quantification of the CCL5 chemokine in clinical samples: the experimental approach consists in the immobilization of anti-CCL5-Biotin capture antibody onto neutravidin-functionalized magnetic microparticles for the selective capture of the target protein followed by conjugation with anti-CCL5 and horseradish peroxidase -labelled IgG for the enzyme-catalyzed amperometric detection onto SPCEs using the H2O2/HQ system.

The work is clearly written, well-structured and all claims of the authors are sustained by experimental evidence. Analytical approach is robust. Innovation with respect to literature is clearly reported. Main important result i.e. the detection of CCL5 at the clinically required levels, is clearly highlighted. Ability to discriminate between patients diagnosed with MS and healthy individuals is a further great result.

 Thank you very much for your kind comments and for your recommendations.

- In the introduction line 72 maybe a typo occurred: “Bearing this in mind and in to provide”. I think “in order to” should be the correct version

 Thanks for this suggestion. We have made the correction in the revised manuscript.

  - Size of figures and tables should fit the template of the journal

 Thanks again, we have modified the size of Figures and Tables as much as possible to fit the template of the Journal

 - Ohm in figure 3 should be same to the unit (Ω) in main text.

 We thank again the Reviewer for this observation.

 ; the underline of “8 oC” (line 429) should be deleted.

 We apologize for this mistake that we have corrected in the revised manuscript.

 In conclusion, as it is a very good job, I would recommend this work for publication in Biosensors in the present form.

 Thank you very much again.

We hope that the current version complies with the high standards and requirements of Biosensors and is now suitable for further consideration and peer review in this Journal.

Thank you very much for your work handling our manuscript. Please do not hesitate to contact us if you need additional information.

Sincerely, 

Prof. Paloma Yánez-Sedeño                                                          

[email protected]

Reviewer 2 Report

A sandwiched immunoassay was designed based on anti-CCL5-Biotin, Neu-MBs, and HPR-IgG to determine the concentration of CCL5 in the biological samples from the healthy people and patients with MS. The working range of 0.1-300 ng/mL and LOD of 40 pg/mL was obtained. I recommend its possible publication in Biosensors for its moving forward the practical application of biochemical sensors for the MS diagnosis. Several comments can be found below:

1.     The dpi of figure 1 and figure 4 should be improved; the caption of figure 4 should be moved below;

2.     The representation of units should be standardized: Ohm in figure 3 should be same to the unit (Ω) in main text; pg mL-1 should be pg/mL or pg·mL-1; the underline of “8 oC” (line 429) should be deleted.

Author Response

Reviewer 2.

A sandwiched immunoassay was designed based on anti-CCL5-Biotin, Neu-MBs, and HPR-IgG to determine the concentration of CCL5 in the biological samples from the healthy people and patients with MS. The working range of 0.1-300 ng/mL and LOD of 40 pg/mL was obtained. I recommend its possible publication in Biosensors for its moving forward the practical application of biochemical sensors for the MS diagnosis.

We greatly appreciate the Reviewer's comments and recommendations.

Several comments can be found below:

  1. The dpi of figure 1 and figure 4 should be improved; the caption of figure 4 should be moved below;

      Thank you very much for your observations. We have corrected these in the new version of the manuscript. 

  1. The representation of units should be standardized: Ohm in figure 3 should be same to the unit (Ω) in main text; pg mL-1should be pg/mL or pg·mL-1; the underline of “8 oC” (line 429) should be deleted.

      We apologize for these mistakes that we have corrected in the revised manuscript.

We hope that the current version complies with the high standards and requirements of Biosensors and is now suitable for further consideration and peer review in this Journal.

Thank you very much for your work handling our manuscript. Please do not hesitate to contact us if you need additional information.

Sincerely,

Prof. Paloma Yáñez-Sedeño

[email protected]